# Language measures correlate with other measures used to study emotion
Shaina Munin [1], Desmond C. Ong[1], Sydney Okland[1], Gili Freedman[2] & Jennifer S. Beer[1] ✉

Researchers are increasingly using language measures to study emotion, yet less is known about whether language relates to other measures often used to study emotion. Building on previous work which focuses on associations between language and self-report, we test associations between language and a broader range of measures (self-report, observer report, facial cues, vocal cues). Furthermore, we examine associations across different dictionaries (LIWC-22, NRC, Lexical Suite, ANEW, VADER) used to estimate valence (i.e., positive versus negative emotion) or discrete emotions (i.e., anger, fear, sadness) in language. Associations were tested in three large, multimodal datasets ($N$s = 193–1856; average word count = 316.7–2782.8). Language consistently related to observer report and consistently related to self-report in two of the three datasets. Statistically significant associations between language and facial cues emerged for language measures of valence but not for language measures of discrete emotions. Language did not consistently show significant associations with vocal cues. Results did not tend to significantly vary across dictionaries. The current research suggests that language measures (in particular, language measures of valence) are correlated with a range of other measures used to study emotion. Therefore, researchers may wish to use language to study emotion when other measures are unavailable or impractical for their research question.

Psychologists use many methods for measuring emotions, including self-report, behavioral coding (e.g., observer ratings, facial cues, vocal cues)[1], and increasingly, analysis of the language that people produce[2,3]. In recent years, there has been more research using language to study emotion amidst improvements in technological resources and the growing amount of available language data on the internet and other sources (e.g., digitized books)[3]. Language methods have facilitated research on a wide range of questions about emotion, such as collective trauma[4–6], consumer behavior[7–9], and emotions in everyday life[10–12]. However, little empirical attention has been paid to understanding whether researchers who select language as their measure of valence (i.e., positive and/or negative emotion) or discrete emotions (e.g., anger, fear, sadness) can consider language to be a likely correlate of other measures used to assess emotion[13–15]. Previous research focuses primarily on associations between language and self-reports[16,17] with less focus on relations to other measures. The present study expands the scope of previous investigations by testing associations between language measures of valence and discrete emotions and other measures that are often used to assess emotion in three large, multimodal datasets.

Researchers have several measurement options available for studying emotion. For example, a long history of research in emotion has used closed-ended self-report scales to assess participants' subjective feelings[18,19]. Other work relies on automated or manual coding of nonverbal behaviors, such as facial or vocal cues[20,21]. In contrast to these other approaches, language methods can be defined as involving the quantitative analysis of natural language, such as the words generated in response to an open-ended prompt (i.e., prompted language) or the words produced in everyday speech or writing (i.e., naturalistic language)[3,22]. As researchers often consider language to reflect attentional focus[23], language measures derived from both prompted language and naturalistic language have been used to assess participants' current affective state[14,24].

Compared to self-report and behavioral coding, language offers at least two benefits for researchers interested in studying emotion. First, researchers can often employ language measures in contexts where self-report or behavioral coding measures are absent. For example, a large body of work has used language methods to assess valence and/or discrete emotions in online spaces, such as social media platforms[5,6,25], consumer review websites[7,9,26], blogs[4], and forums[27]. Researchers also use language to estimate valence and/or discrete emotions in text sources such as written diaries[12,17] and historical text corpora[28,29]. Moreover, researchers leveraging archival datasets that lack self-report or behavioral measures of emotion may use language to measure valence and/or discrete emotions[15].

[1]Department of Psychology, The University of Texas at Austin, Austin, TX, USA. [2]Department of Psychology, St. Mary's College of Maryland, St. Mary's City, MD, USA. ✉e-mail: beerutexas@gmail.com

Second, language measures tend to be unintrusive and easily implemented[30] and therefore hold the potential to circumvent limitations of other measures. For instance, self-reporting emotion may be challenging over a long timeframe. Participants may overemphasize or underemphasize emotions which fluctuate across a long time when asked for a single retrospective report of their emotion[31,32] or another person's emotion[33]. Yet repeated 'real time' self-report measurements aiming to track emotional fluctuations over a long duration may dampen participants' emotion ratings[34]. Analysis of naturalistic language may therefore provide a promising alternative to self-report when assessing valence and/or discrete emotions over long timeframes. Furthermore, some researchers may lack access to resources to implement self-report or behavioral coding in their work. Collecting self-report or behavioral coding may require participant funds and/or time, lab space, or large teams of research assistants that may not be available. By contrast, research using language can be conducted with little cost beyond a computer and Wi-Fi access using public datasets and/or text scraped from the internet. Thus, automated language analysis can bypass self-report or resource intensive approaches like behavioral coding[30].

How do researchers tend to operationalize language when using language to study emotion? A common approach is to use pre-made word lists called dictionaries to identify words within a given text that directly (e.g., "happy", "sadness") and/or indirectly (e.g., "good", "funeral") relate to emotion[23,30]. Dictionaries which focus on valence classify emotion-related words by whether they can be presumed to reflect positive and/or negative emotion[35]. Some dictionaries additionally classify emotion-related words by whether they can be presumed to reflect a discrete emotion, often focusing on discrete negative emotions (e.g., anger, fear, sadness)[36].

Many dictionaries have been developed to measure emotion-related words[37], and they tend to fall into three general categories: word counting, word weighting, and rule-based dictionaries. Word counting dictionaries such as Linguistic Inquiry and Word Count (LIWC: Pennebaker et al.[38]) and the NRC Emotion Lexicon[39,40] measure the relative frequency of emotion-related word use[23]. On the other hand, word weighting dictionaries such as the NRC Emotion Intensity Lexicon[41], the Lexical Suite[26,35], and the Affective Norms of English Words[42] differentiate emotion-related words rather than counting each word equally. For example, a word counting dictionary would not differentiate between words representing different degrees of emotional intensity (e.g., "annoyed" and "furious") within the same category (e.g., anger words), whereas a word weighting dictionary would assign a higher anger score to "furious" than "annoyed." Finally, rule-based dictionaries such as VADER[43] extend word counting and weighting dictionaries by modifying emotion scores based on contextual factors (e.g., punctuation, qualifiers; "hardly annoyed" versus "very annoyed!").

The relations between language operationalizations and other measures often used to study emotion have been largely untested, and the few previous investigations are somewhat constrained in their scope. Some previous research has investigated the associations between language and self-reports of emotion and yielded a mix of significant and non-significant findings. For example, one study found significant associations between spoken language following a film task and self-ratings of emotion (i.e., the PANAS[18]) for positive valence and amusement, but not for negative valence and sadness[24]. Other work found significant correlations between language and self-reports of valence and discrete emotions (i.e., anger, anxiety, sadness) for prompted spoken language, but not for naturalistic written language (i.e., text messages)[44]. Another study found significant associations between written diary language and self-ratings of positive valence for daily assessments, but not for weekly assessments[17]. And although one recent study found significant associations between self-reports of valence and estimates of valence from participants' written descriptions of their emotions[45], other studies did not find any significant associations between self-reports of valence and estimates of valence from either naturalistic spoken language[46] or social media posts[16].

Less work has examined associations between language and measures other than self-reported emotion. For example, research in computational linguistics finds that language measures are associated with observer ratings

of the writer's valence in online written posts[37,47]. Yet very few studies have tested associations with observer reports in offline contexts where other signals beyond language are available. For example, one small study (N = 20) found significant associations between language and observational coding for negative emotion, but not for positive emotion[48], and another study did not find consistent associations between language and listeners' valence ratings of audio clips[46]. To date, however, no previous work has tested whether language measures of valence and/or discrete emotions are associated with facial cues or vocal measures. Furthermore, in psychology publications, most previous work has focused on LIWC[16,46], and less is known about the associations of other dictionary measures (e.g., word weighting and rule-based approaches). Therefore, research is needed to test associations between a wider range of language dictionaries and other measures often used to study emotion.

To better understand how language relates to other measures often used to study emotion, the present study tests associations between a broader range of dictionaries (i.e., LIWC-22, NRC, Lexical Suite, ANEW, VADER) and a broader range of measures (i.e., self-report, observer report, facial cues, and vocal cues), compared to previous work. We test associations in 3 large, multimodal datasets (see Table 1). Dataset 1 draws from personal narratives on a specific negative topic (i.e., social rejection). Analyses of Dataset 2 explore whether associations generalize to a diverse sample of positive and negative narratives. Finally, analyses of Dataset 3 (3a and 3b) build on the brief lab-based tasks in Datasets 1 and 2 by examining whether associations between language and other measures used to study emotion emerge in naturalistic interpersonal interactions.

## Methods
### Datasets
**Dataset 1: social rejector narratives**. Dataset 1[49] consisted of written social rejector narratives from 602 participants based in the southern United States (see Table 1). Participants were brought into the lab and instructed to write about any time in their life when they socially rejected another person. Social rejection was defined as denying someone's request for social inclusion or acceptance (e.g., denying a first date request, ending a friendship). Participants were asked to write everything they remembered about the rejection, including the request they rejected, when the rejection took place, and their relationship with the rejected person. Each participant recalled one social rejection for a total $N = 602$ narratives ($M_{length} = 316.7$ words, $SD_{length} = 210.6$ words). Data was collected with approval from the UT Austin Institutional Review Board. Informed consent was obtained from all participants; participants received either course credit or monetary compensation for participating in the study.

**Dataset 2: SEND narratives**. Dataset 2 was drawn from the Stanford Emotional Narratives Dataset (SEND)[50], a publicly accessible dataset of spoken emotional narratives from 49 participants based in the West Coast of the United States (see Table 1). Participants were brought into the lab and instructed to recall the three most positive and negative events they felt comfortable sharing in front of a camera. Participants talked for as long as they wanted about the event ($M_{duration} = 135$ seconds, $SD_{duration} = 41$). A subset of these narratives that met inclusion criteria (i.e., target's face always in frame, clear emotional narrative, no sensitive content) comprised the SEND[51], resulting in $N = 193$ video transcripts ($M_{length} = 461.9$ words, $SD_{length} = 315.6$ words).

**Dataset 3 (3a & 3b): CANDOR conversations**. Datasets 3a and 3b were drawn from the CANDOR corpus[52], a publicly accessible dataset of naturalistic dyadic conversations from 1456 participants who were based in the United States (see Table 1). Participants took part in a video conversation with a randomly matched stranger between January and November 2020. Participants were instructed to have a "get to know you" conversation for at least 25 minutes ($M_{duration} = 31.3$ minutes, $SD_{duration} = 7.96$). Notably, over half of participants returned for one or

**Table 1 | Overview of datasets in the present study**

| | Dataset 1[49] | Dataset 2[50] | Dataset 3a[52] | Dataset 3b[52] |
|---|---|---|---|---|
| # Texts | 602 | 193 | 1455 | 1856 |
| Mean # Words (SD) | 316.7 (210.6) | 479.0 (325.7) | 2776.0 (1173.5) | 2782.8 (1092.0) |
| # Participants | 602 | 49 | 1455 | 754 |
| Mean Age (SD) | 25.6 (11.9) | 23.7 (7.9) | 33.2 (11.1) | 34.6 (11.5) |
| % Women | 70.4 | 55.1 | 53.6 | 55.0 |
| % Men | 26.9 | 44.9 | 41.7 | 41.6 |
| % Other Gender | 2.2 | 0 | 1.8 | 2.1 |
| % White | 39.9 | 32.7 | 62.6 | 63.5 |
| % Asian | 22.8 | 18.4 | 13.9 | 13.9 |
| % Black | 3.3 | 4.1 | 7.8 | 8.5 |
| % Hispanic/ Latino | 19.9 | 8.2 | 7.4 | 6.4 |
| % Other or Mixed | 14.0 | 36.7 | 5.2 | 6.2 |
| Modality | Written | Spoken | Spoken | Spoken |
| Context | Monologue | Monologue | Dialogue | Dialogue |
| Topic | Social rejector narratives | Positive and negative narratives | Getting-to-know-you | Getting-to-know-you |
| Self-Report | X | X | X | X |
| Observer Report | | X | | |
| Facial Cues | | X | X | X |
| Vocal Cues | | X | X | X |

Gender identity and race/ethnicity were self-reported by participants in each dataset.

**Table 2 | Language measures used in the present study**

| Dictionary | Measure | Description |
|---|---|---|
| LIWC-22 | Tone | Standardized percentile score (0 to 100) based on the proportion of positive minus negative words |
| | Anger | Proportion of anger-related words |
| | Anxiety | Proportion of anxiety-related words |
| | Sadness | Proportion of sadness-related words |
| NRC | Pos – Neg | Proportion of positive minus negative words |
| | Anger | Proportion of anger-related words |
| | Fear | Proportion of fear-related words |
| | Sadness | Proportion of sadness-related words |
| | Anger Intensity | Average intensity score (0 to 1) of anger-related words |
| | Fear Intensity | Average intensity score (0 to 1) of fear-related words |
| | Sadness Intensity | Average intensity score (0 to 1) of sadness-related words |
| Lexical Suite | Valence | Average valence score (0 to 9) |
| ANEW | Valence | Average valence score (1 to 9) |
| VADER | Compound | Normalized valence composite score (-1 to 1) |
| | Pos – Neg | Positive minus negative intensity score |

more additional conversations with new, randomly matched conversation partners. Thus, the corpus consists of 1656 conversations (i.e., 3312 individual speaker transcripts) from 1456 unique participants.

To leverage the size and participant composition of the CANDOR corpus, the present analyses tested replication in two distinct subsamples (Datasets 3a and 3b). Dataset 3a consisted of individual speaker transcripts from the first time participants engaged in a conversation as part of the study. One transcript was removed from the subsample due to low word count, resulting in a final $N = 1455$ transcripts ($M_{length} = 2522.0$ words, $SD_{length} = 1161.4$ words) from 1455 unique participants in Dataset 3a. Dataset 3b consisted of individual speaker transcripts from the additional conversations engaged in by the participants who returned to the study one or more times, resulting in $N = 1856$ transcripts ($M_{length} = 2544.6$ words, $SD_{length} = 1086.5$ words) from 754 unique participants.

### Reporting summary
Further information on research design is available in the Nature Portfolio Reporting Summary linked to this article.

## Measures
### Language
Fifteen language measures were drawn from 5 dictionaries (LIWC-22, Lexical Suite, NRC, VADER, ANEW; see Table 2). Six of these measures (LIWC-22 Tone, NRC Pos – Neg, Lexical Suite Valence, ANEW Valence, VADER Compound, and VADER Pos – Neg) measure words relating to valence. Higher scores on each measure reflect more positively and less negatively valenced language. The remaining 9 measures (from LIWC-22 and NRC) measure words relating to discrete emotions. We focus on anger, fear, and sadness as these are the only discrete emotions included in LIWC-22[36].

One language measure (VADER Compound) was dropped from analyses of Datasets 3a and 3b due to a restriction in variance arising from its calculation. That is, VADER's Compound score is derived from the sum of word-level scores and therefore texts with more words are more likely to produce normalized values on the extremes (i.e., −1 and 1). For example, 99.2% of texts in Dataset 3 (~2800 words each) had a Compound score of 1, compared to 4.7% of texts in Dataset 2 (~480 words each). Therefore, we include an additional measure of valence from VADER (Pos – Neg) which is not subject to the same variance issues as the Compound measure.

### Self-report
Self-reports of emotion were measured in all three datasets. In Dataset 1, participants responded to 16 items about discrete emotions they remembered experiencing during the recalled rejection on a 1 (*not at all*) to 5 (*extremely*) scale. A combined rating of positive minus negative emotions[53] was computed from the average rating of 3 positive emotions (happy, warm/friendly, enjoyment) minus the average of 13 negative emotions (impatient, drained, frustrated/annoyed, depressed/sad, angry/hostile, worried/anxious, guilty, ashamed, embarrassed, resentful, self-conscious, hurt, and numb).

In contrast to the discrete emotion items in Dataset 1, Datasets 2 and 3 used self-report scales which ranged from negative to positive. In Dataset 2, participants watched each video they recorded and rated their emotional experience in the video at each 0.5 s on a 0 (*very negative*) to 100 (*very positive*) scale. In Dataset 3, participants provided self-reports of their emotion at three timepoints: a momentary account of their emotion before the conversation, a momentary account of their emotion after the conversation, and a retrospective account of how they had felt during the conversation (reported immediately after the conversation). Each was rated on a 1 (*extremely negative*) to 9 (*extremely positive*) scale.

### Observer report
Observer ratings of emotion were measured in Dataset 2. As part of the data collection procedure for Dataset 2, additional participants ($N = 700$) were

**Fig. 1 | Forest plot of selected Spearman correlations between language measures of valence and self-report, observer report, facial cues, and vocal cues.** Self-report measure for Datasets 3a and 3b is the retrospective account of emotion across the task. Error bars indicate 95% confidence intervals. Dataset 1 Ns = 588–602, Dataset 2 Ns = 193, Dataset 3a Ns = 1411–1455, Dataset 3b Ns = 1833–1856. For other correlations, see Supplementary Tables S4a–S4d and S6a–S6d.

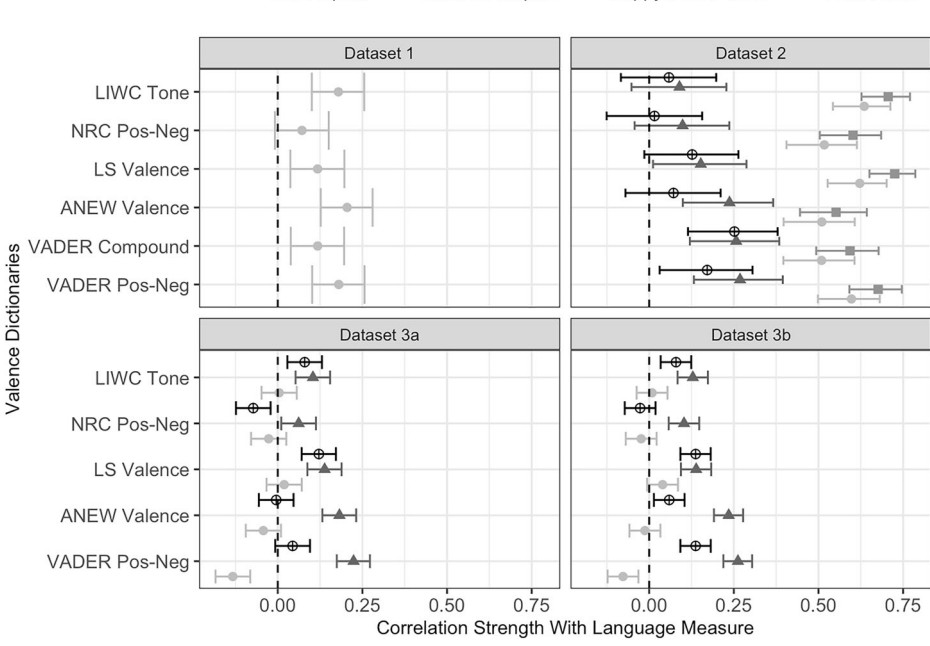

recruited on Amazon Mechanical Turk to provide observer ratings[51]. Observers rated the speaker's emotional experience in each video at each 0.5 second on a 0 (*very negative*) to 100 (*very positive*) scale.

**Facial cues**

Facial cues were measured in Datasets 2 and 3. In Dataset 2, an automated software program (Emotient by iMotions) extracted moment-to-moment measures of the likelihood of different facial expressions (e.g., happy, angry, fearful, sad) for the speaker in each video (the original paper additionally used the software to extract the presence of 20 Facial Action Units)[52]. In Dataset 3, a convolutional neural network trained on a facial image database extracted moment-to-moment measures of the likelihood of different facial expressions (e.g., happy, angry, fearful, sad) for each speaker in each conversation[54].

**Vocal cues**

Vocal cues were measured in Datasets 2 and 3. In Dataset 2, an automated software program (openSMILE v2.3.0) extracted moment-to-moment extended GeMAPS (eGeMAPS) parameter measures of vocal pitch (i.e., fundamental frequency, F0) and vocal intensity (i.e., loudness) for the speaker in each video[51]. In Dataset 3, an automated software package (Parselmouth) extracted moment-to-moment measures of vocal pitch (i.e., F0) for each speaker in each conversation. Additionally, a model trained on an emotional speech dataset estimated moment-to-moment measures of vocal emotional intensity for each speaker in each conversation[54].

**Analysis plan**

Data were cleaned in two primary ways. First, word counts from the NRC measures were converted to proportions of text to account for the total number of words. LIWC-22 performs this correction automatically[36], and the Lexical Suite, ANEW, and VADER measures do not use word counts. Second, moment-to-moment self-report, observer-report, facial, and vocal measures in Datasets 2 and 3 were averaged across the whole video to test for between-person, rather than within-person, associations with language.

As several of the language and non-language measures were skewed (i.e., |skewness| >1), associations between language and other signals of emotion were tested using Spearman correlations[55]. Missing data were removed pairwise. Correlation analyses were conducted in R (v. 4.3.2) using the Hmisc[56] and DescTools[57] packages. Analyses were not pre-registered. For descriptive statistics, see Supplementary Tables S1a-S2.

## Results

To account for the number of comparisons tested between each set of measures, we report Bonferroni-corrected results throughout. For full results, including original *p*-values, corrected *p*-values, and 95% confidence intervals, see Supplementary Tables S4a-S4d and S6a-S6d.

### Language measures consistently related to self-report in two of the three datasets

Language measures of valence and discrete emotions significantly related to self-report in Datasets 1 and 2. Of the 6 language measures of valence in analyses of Datasets 1 and 2, 5 were significantly associated with more positive (relative to negative) self-reported emotion in Dataset 1 ($\rho = 0.07$ to 0.20), and all 6 were significantly associated with more positive self-reports in Dataset 2 ($\rho = 0.51$ to 0.64) (see Fig. 1). Similarly, of the 9 language measures of discrete negative emotions (anger, fear, sadness), 6 were significantly associated with more negative (relative to positive) self-reported emotion in Dataset 1 ($\rho = -0.18$ to $-0.02$), and 8 were significantly associated with more negative self-reports in Dataset 2 ($\rho = -0.43$ to $-0.03$). In addition, 8 of the 9 language measures of anger, fear, or sadness were significantly associated with greater self-reported frustration, worry, or hurt, respectively, in Dataset 1 ($\rho = 0.05$ to 0.29; see Supplementary Table S6a).

However, no significant associations between language measures and self-report replicated across both Datasets 3a and 3b. Of the 5 language measures of valence in analyses of Datasets 3a and 3b, none were significantly associated with more positive pre-task self-reported emotion (Dataset 3a: $\rho = -0.10$ to 0.07; Dataset 3b: $\rho = -0.07$ to 0.05), more positive retrospective self-reports (Dataset 3a: $\rho = -0.13$ to 0.02; Dataset 3b: $\rho = -0.08$ to 0.04), or more positive post-task self-reports (Dataset 3a: $\rho = -0.14$ to 0.01; Dataset 3b: $\rho = -0.05$ to 0.06) in both datasets (see Fig. 1). Similarly, none of the 9 language measures of discrete negative emotions were significantly associated with more negative pre-task self-reported emotion (Dataset 3a: $\rho = -0.06$ to 0.02; Dataset 3b: $\rho = -0.09$ to $-0.02$), more negative retrospective self-reports (Dataset 3a: $\rho = 0.00$ to 0.05; Dataset 3b: $\rho = -0.01$ to 0.06), or more negative post-task self-reports (Dataset 3a: $\rho = 0.01$ to .06; Dataset 3b: $\rho = -0.02$ to 0.05) in both datasets.

## Language measures consistently related to observer report in one dataset

Language measures of valence and discrete emotions significantly related to observer reports. All 6 language measures of valence were significantly associated with more positive observer-reports of emotion in Dataset 2 ($\rho = 0.55$ to $0.73$) (see Fig. 1). Additionally, 8 of the 9 language measures of discrete negative emotions were significantly associated with more negative observer-reported emotion in Dataset 2 ($\rho = -0.48$ to $0.06$).

## Language measures varied in their associations with facial cues in two datasets

Language measures of valence significantly related to facial cues, but language measures of discrete emotions generally did not show significant evidence of associations with facial cues. In Dataset 2, 3 of the 6 language measures of valence were significantly associated with greater likelihoods of happy faces ($\rho = 0.09$ to $0.27$). Four of the 5 of the language measures of valence in analyses of Datasets 3a and 3b were also significantly associated with greater likelihoods of happy faces in both datasets (Dataset 3a: $\rho = 0.06$ to $.22$; Dataset 3b: $\rho = 0.10$ to $0.26$) (see Fig. 1). Additionally, 2 language measures of valence were significantly associated with lower likelihoods of negative faces (i.e., average of angry, fearful, and sad facial cues) in Dataset 2 ($\rho = -0.28$ to $0.01$), and 3 language measures of valence were significantly associated with lower likelihoods of negative faces in both Datasets 3a ($\rho = -0.25$ to $-0.05$) and 3b ($\rho = -0.27$ to $-0.06$). For individual correlations with angry, fearful, and sad facial cues, see Supplementary Table S4c.

However, language measures of discrete emotions did not tend to show significant associations with facial cues. Of the 9 language measures of anger, fear, or sadness, none were significantly associated with greater likelihoods of angry, fearful, or sad faces in Dataset 2 ($\rho = -0.08$ to $0.15$), and only 1 was significantly associated with greater likelihoods of angry, fearful, or sad faces in both Datasets 3a ($\rho = -0.04$ to $0.07$) and 3b ($\rho = -0.03$ to $0.06$).

## Little evidence for associations between language measures and vocal cues in two datasets

Although a few language measures of valence and discrete emotions significantly related to vocal pitch and vocal intensity, results tended to be inconsistent. One of the 6 language measures of valence in Dataset 2 was significantly associated with greater vocal pitch ($\rho = 0.02$ to $0.25$), and 2 of the 5 language measures of valence in analyses of Datasets 3a and 3b were significantly associated with greater vocal pitch in both datasets (Dataset 3a: $\rho = -0.07$ to $0.12$; Dataset 3b: $\rho = -0.03$ to $0.14$) (see Fig. 1). None of the language measures of valence were significantly associated with vocal intensity in Dataset 2 ($\rho = -0.06$ to $0.11$) or in both Datasets 3a and 3b (Dataset 3a: $\rho = -0.06$ to $0.05$; Dataset 3b: $\rho = -0.04$ to $0.05$).

Furthermore, in Dataset 2, 1 of the 9 language measures of discrete negative emotions was significantly associated with lower vocal pitch ($\rho = -0.21$ to $0.12$), and none were significantly associated with vocal intensity ($\rho = -0.15$ to $0.06$). However, the reverse direction of associations emerged in Datasets 3a and 3b. That is, 4 language measures of discrete negative emotions were associated with greater vocal pitch (Dataset 3a: $\rho = -0.02$ to $0.15$; Dataset 3b: $\rho = 0.00$ to $0.19$), and 2 were associated with greater vocal intensity (Dataset 3a: $\rho = -0.01$ to $0.14$; Dataset 3b: $\rho = -0.01$ to $0.18$) in both Datasets 3a and 3b.

## Little evidence for variations in associations across different language dictionaries

Lastly, the patterns of associations across datasets generally did not significantly vary among the language dictionaries. Ten paired t-tests tested differences in the patterns of associations between 5 language measures of valence (excluding VADER Compound for incomplete data) and found only 2 significant differences (between NRC-LIWC and NRC-Lexical Suite; see Supplementary Table S7a). Furthermore, Spearman correlations between the patterns of associations for language measures of valence all exceeded 0.60. Correlations involving the NRC measure of valence ($\rho = 0.64$ to $0.74$) were lower than correlations not involving the NRC

($\rho = 0.81$ to $0.97$). Thus, all language measures of valence showed similar patterns of associations with other measures, but the NRC measure of valence may be least congruent with the other language measures.

The LIWC and NRC measures of discrete emotions also tended not to significantly differ in their patterns of associations across datasets. Across 9 paired t-tests testing differences in the patterns of associations between the 3 language measures of anger, 3 language measures of fear, and 3 language measures of sadness, respectively, only 1 significant difference emerged (between LIWC Anxiety and NRC Fear Intensity; see Supplementary Table S8).

## Discussion

Researchers who want to study emotion but cannot use measures often used to assess emotion are increasingly turning to language[13–15], yet limited previous work has tested associations between language and other measures. The present study examined 3 large, multimodal datasets to test whether language measures of valence and discrete emotions were associated with self-report, observer report, facial cues, and vocal measures. Language measures showed largely consistent patterns of significant associations with self-report and observer report. Language measures of valence also showed significant associations with facial cues, but language measures of discrete emotions did not. Moreover, the selection of a particular dictionary tended to not make a difference; patterns of associations across different dictionaries were highly correlated with one another. The current findings are consistent with the idea that language measures may be a useful option when self-report and behavioral coding measures are unavailable or impractical for a particular research question.

### Language showed significant evidence of associations with self- and observer report but less consistent evidence of associations with facial and vocal measures

The present findings suggest that the associations between language and other measures used to study emotion may depend on the other measures considered, and in some cases, the language measures involved. For example, language dictionaries showed a robust pattern of significant associations with observer report and with self-report in two of the three datasets. Associations with self- and observer report were consistent across both language measures of valence and discrete emotions. These findings align with some previous studies which found significant correlations between language measures and self-reports[17,44,45] and observer reports[48]. Furthermore, the present work extends prior findings across a range of dictionary measures.

However, significant evidence for associations between language and facial and vocal cues was less consistent. When facial cues were considered, language measures of valence showed patterns of significant associations, yet language measures of discrete emotions did not. There was no statistically significant evidence of a relationship between language and vocal measures across the datasets. The present findings may contribute to ongoing discussion about the extent to which nonverbal cues relate to measures of valence or discrete emotions. That is, some researchers argue that facial cues do not reliably map onto measures of discrete emotions[58]. Moreover, while previous research suggests that vocal pitch and vocal intensity are associated with discrete emotions[59], vocal cues may be more strongly related to physiological arousal than valence[60]. Inconsistent associations between nonverbal cues and language measures of valence and discrete emotions may be considered consistent with these views.

Taken together, the current findings suggest that language may better approximate self-report and observer reports than vocal measures. The present work also raises the possibility that associations between language and facial cues may be stronger for dictionaries which operationalize valence compared to dictionaries which operationalize discrete emotions.

### Are language dictionaries less susceptible to display rules?

A number of the dictionaries considered in the present study did not tend to show significant associations with facial cues but did show associations with

other measures. In addition to the perspectives on variable associations between discrete emotions and facial cues discussed above, another possible explanation for the inconsistent associations with facial cues may be the operation of display rules[61]. That is, there may have been reduced association due to norms against facial displays of negative feelings, whereas negative feelings may have seeped into language relatively less unchecked. Consistent with the possibility that suppression of negative facial displays may underlie the lack of association with language, previous work suggests that correlations between measures of emotion diminish when facial movements are suppressed[62]. Indeed, in Dataset 2, facial cues of anger, fear, and sadness tended to be weaker predictors of self-reports of emotion than language measures were (see Supplementary Table S3b), suggesting that facial cues may have less reliably tracked with negative feelings. In Datasets 3a and 3b, facial cues tended not to significantly relate to self-report in the expected direction, though it is unclear whether the lack of association may stem from issues with the self-report measures (see below).

### Does timing shape when language rather than self-report may be most useful?

Why did language measures not significantly relate to self-reported emotion in a 30-minute conversation (Datasets 3a and 3b) but did significantly relate to self-report in a 2-minute monologue task (Dataset 2)? One possibility is that the current findings lend credence to concerns about the limitations of self-report when measuring emotion over long periods of time and support the preference for language measures under those circumstances. Researchers may find it impractical or disruptive to have participants continuously self-report their emotion over a 30 min task[34]. However, self-reports at other timepoints (i.e., retrospective, pre-task, post-task) may unreliably approximate measures of emotion collected across the 30 min duration. For example, retrospective self-reports of emotion are often disproportionately colored by individuals' peak and ending emotions[31,63]. Momentary pre-task and post-task self-reports may similarly fail to tap into participants' emotions throughout a task[64]. Therefore, language may be preferable over self-reports when researchers seek to measure emotion over longer timescales.

However, another interpretation is that the present findings are consistent with some prior work suggesting that self-reported emotion may relate less strongly to language drawn from naturalistic settings than language generated in response to an experimental prompt. For example, language does not tend to significantly relate to self-report when sampled from text messages, social media posts, or everyday spoken language[16,44,46]. Yet language has been found to significantly relate to self-report when participants are prompted to write or speak about an emotional experience[24,44,45]. In the present study, the monologue about an emotional event (Dataset 2) more closely resembles the prompted context, whereas the "getting-to-know-you" conversation (Datasets 3a and 3b) may be more similar to a naturalistic setting. Thus, future research may wish to more systematically test whether naturalistic or prompted contexts may influence associations between language and self-report.

### Social context may play a role in the coherence between language, facial, and vocal measures

The current findings are also consistent with recent suggestions that coherence across behavioral measures of emotion may differ between inter- and intrapersonal contexts. For example, a greater number of significant associations emerged between language measures and both facial cues and vocal pitch in dyadic conversations (Datasets 3a and 3b) than in spoken monologues to a camera (Dataset 2). Some theories of emotion contend that coherence across emotional response systems should emerge regardless of context[65,66], yet evidence in support of emotional coherence is mixed[67]. More recent theoretical developments suggest that emotional expressions function as communicative acts[68–70]. When individuals engage in an interaction with another person, they may use interpersonal feedback to modulate their nonverbal and vocal behavior to achieve a communicative goal[68,71]. Therefore, it is possible that enhanced behavioral coordination due to

immediate feedback in live interpersonal interactions may allow for stronger associations between measures of emotion compared to imagined interactions elicited when alone or speaking to a camera[72].

That said, the greater number of texts and words in the conversation task (Datasets 3a and 3b) relative to the monologue task (Dataset 2) may also have afforded greater power to detect significant associations between language and behavioral measures of emotion. Furthermore, the rapid advancement of machine learning algorithms to extract emotion measures from video data[73] raises the possibility that the algorithms which quantified the facial and vocal measures in the monologue task may have been less precise than the more recently developed models which quantified these measures in the conversation task. Thus, future work is needed to understand whether interpersonal feedback is important for observing coherence across emotion measures.

### Limitations

The present study design has a number of strengths, including the broad consideration of dictionaries, emotion measures, and contexts through analyses of multiple datasets; however, there are limitations to the present work as well. First, all analyses in the present study were conducted with American English-speaking samples. English is known to be a linguistic and cultural outlier among world languages[74], and most prior work on language and emotion only considers English speakers[3]. Still, it is possible that associations between language and other measures used to study emotion may generalize to other languages beyond English. For example, ratings of positive valence for words in one language tend to remain stable when the words are translated into other languages[75]. Additionally, some previous work finds that correlations among measures of emotion vary little between English-speaking and non-English-speaking cultures[76]. Future research with non-English-speaking samples is ultimately needed to test whether the present results indeed replicate across languages. Of the dictionaries tested in the present study, LIWC has been translated into at least 14 languages[36], and the NRC Emotion Lexicon has been translated into over 100 languages[3], which can facilitate tests in non-English contexts.

Second, the present study tested whether different measurement approaches to emotion relate to one another but cannot speak to the degree to which these measures truly reflect emotional content. Several theories of emotion argue that measures of emotion should relate to one another because they reflect components of an emotional response[65,66]. Still, it is possible that these measures could also tap into other processes. For example, emotion-related language could be used for purposes other than expressing one's current emotion (e.g., "she looks happy")[47], and changes in facial behavior could be motivated by situational demands rather than emotional impulses (e.g., smiling to be polite). Further research is necessary to inform broader questions about the associations between measures of emotion and the emotional content itself.

Third, the present study relied on automated machine learning approaches to measure facial and vocal cues, which may not always identify cues with high accuracy. For example, research suggests that automated approaches to measuring facial cues can perform well under controlled conditions yet show significantly greater error when applied to real-world stimuli[77] or even when generalized to other lab-based datasets[78]. In the present study, the model used to identify facial cues in the conversation dataset (Dataset 3) was not adapted to naturalistic conversation contexts, where facial movement tends to differ considerably from the prototypical poses used in training datasets[54]. Moreover, the two datasets in the present work which included automated measurements of facial and vocal cues (Datasets 2 and 3) used different software programs, which could have exacerbated variability in measurement across the two contexts. Therefore, future research with naturalistic data may wish to use a standardized machine learning approach and complement such an approach with human-coded behavior to provide a more complete picture of facial and vocal cues which may relate to language approaches to studying emotion.

## Conclusions

The increasing use of language to study emotion[3] presents an opportunity to (a) facilitate new research in contexts and datasets which lack other measures often used to study emotion and (b) circumvent the perceived limitations and costs associated with other measures. Our work highlights that language measures (and in particular, language measures of valence) show consistent associations with other measures often used to assess emotion. Associations did not tend to significantly vary across different dictionaries used by researchers to operationalize valence or discrete emotions in language. The findings may also be considered consistent with concerns that self-report may inadequately measure emotions over long timescales and raise the possibility that language measures may be preferable in such circumstances. Additionally, the findings are consistent with speculations that coherence across measures of emotion may be enhanced in live interpersonal interactions. Overall, the present work supports the idea that researchers who are unable to utilize self-report or behavioral coding may find language to be a useful measure in their work.

## Data availability

The three datasets used in the present work are publicly available (Dataset 1: https://osf.io/v2wpd[49]; Dataset 2: https://github.com/StanfordSocialNeuroscienceLab/SEND[50]; Dataset 3: https://betterup-data-requests.herokuapp.com/[52]).

## Code availability

Analysis code can be publicly accessed at https://osf.io/zw3e5[79].

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

## Acknowledgements

The data collected for Dataset 1 was supported by a grant awarded to G.F. and J.B. from the National Science Foundation (BCS-2017085). The funders had no role in study design, data collection and analysis, decision to publish or preparation of the manuscript. Any opinion, findings, conclusions, or recommendations expressed in this material are those of the authors and do not necessarily reflect the views of the National Science Foundation.

## Author contributions

Conceptualization by S.M. and J.B.; Methodology by S.M., D.O., and J.B.; Investigation by S.O.; Formal Analysis by S.M.; Visualization by S.M. and D.O.; Writing-Original Draft by S.M.; Writing-Review & Editing by S.M., D.O., S.O., G.F., and J.B.; Supervision by D.O., G.F., and J.B.; Funding Acquisition by G.F. and J.B.

## Competing interests

The authors declare no competing interests.
