## [Transparent Peer Review file · Communications Psychology]

Language Measures Correlate With Other Measures Used to Study Emotion

Corresponding Author: Ms Shaina Munin

Version 0:

Decision Letter: first round

Dear Ms Munin,

Thank you for your patience during the peer-review process. Your manuscript titled "Self-report, Behavior, and Emotion Words: Examining Associations Between Language and Other Measures Used to Study Emotion" has now been seen by 2 reviewers, and I include their comments at the end of this message. They find your work of interest but raised some important points. We are interested in the possibility of publishing your study in Communications Psychology, but would like to consider your responses to these concerns and assess a revised manuscript before we make a final decision on publication.

We therefore invite you to revise and resubmit your manuscript, along with a point-by-point response to the reviewers. Please highlight all changes in the manuscript text file.

Editorially, we consider that you should clarify terminology throughout, such as by using more precise language for what is being assessed under "language" and "emotion-related words". Additionally, it would be helpful to make a clearer distinction between language and self-report, given their overlap, and to specify whether the underlying emotional content remains consistent across the different measures (e.g., language, self-report, facial expressions, and vocal expressions). Please also ensure you address the concerns regarding the methodological details of the studies.

Please ensure you follow our statistical guidelines when reporting statistics (<https://www.nature.com/commspsychol/submit/submission-guidelines#statistical-guidelines>). Please note in particular our requirements for the reporting and interpretation of null-results. Non-significant findings derived from null-hypotheses significance tests should be reported in full, but may not be interpreted. Where you interpret null results, this interpretation must be based on Bayes Factors or equivalence tests.

I am attaching an Editorial Requests Table that details critical reporting requirements for the revised manuscript. Please attend to each item and ensure your manuscript is fully compliant. We are requesting that your manuscript aligns with these requirements as this facilitates the evaluation of your manuscript, reducing delays in re-review and potential future acceptance. If your revised manuscript is not aligned with these requests on major issues, such as those concerning statistics, it may be returned to you for further revisions without re-review. Additional information can be found in our style and formatting guide <https://www.nature.com/documents/commspsychol-style-formatting-guide-accept.pdf> >Communications Psychology formatting guide.

Please use the following link to submit your

- revised manuscript,
- point-by-point response to the referees' comments,
- cover letter (as a separate document),
- the Editorial Policy Checklist (see below),
- the Reporting Summary (see below), and
- the completed Editorial Request Table (attached):

Link Redacted

Best regards,

Yafeng Pan

Yafeng Pan, PhD
Editorial Board Member
Communications Psychology
orcid.org/0000-0002-5633-8313

REVIEWER EXPERTISE:

Reviewer #1: Emotion and language

Reviewer #2: Emotional measures

REVIEWER REPORTS:

Reviewer #1 (Remarks to the Author):

In this paper, the authors compare methods of assessing emotion across multiple modalities (natural language, self-report, other-report, facial movements, vocal features). These comparisons are repeated across multiple datasets that themselves vary in modality (spoken, written) and context (monologic, dialogic), which is a real asset to this work and raise interesting questions about why some relationships emerge in some contexts but not others. The results provide a necessary descriptive basis for orienting researchers interested in using natural language for the assessment of emotion, broadly construed.

I have the following comments and questions for the authors, aimed at clarifying the terminology throughout, qualifying their findings, and extending the implications of this work to a broader range of readers.

1. I recommend more precise wording for what is being assessed in language. Valence is not the same as emotion; it is one (critical) dimension of emotion. Valence does not capture the richness of emotional experience; language-estimated valence is also not perfectly correlated with self-report, even when the latter is maximally reliable. It seems more felicitous to refer to 'affect' as the broader construct being estimated in language, or even simply to refer to valence. This is especially so given that valence was the only self-/other-report measure collected in Datasets 2 and 3.

2. I disprefer the phrase 'facial expression' and the use of emotion category labels (e.g., 'happy') for them, when what's really coded for is combinations of facial movements that are often associated with emotion categories. In the methods, the authors describe their facial coding algorithms as estimating the likelihood of particular facial expressions, which I don't mind because it explicitly states that these are likelihoods. Overall, I find 'cues' (which is already in use) to be a more theory-neutral term.

3. There are several recent papers that the authors could incorporate into their literature review and discussion. For example, Carlier et al (2021) investigated the relationship between self-report and estimates of valence and emotion across multiple modalities (e.g., spoken natural language, vocal acoustics, text messages), finding weak correlations overall and modest results for valence and happiness. Hoemann et al (2024) found that labels for current experience recapitulated valence ratings.

4. I wasn't sure how to feel about the argument that repeated real-time self-reports may impact the emotional process (page 5). I do not know what the 'emotional process' is, or how we can get away without impacting it whenever we explicitly ask participants to reflect and respond in ways and at times they might not have otherwise. More critically, this argument seems to constrain the types or sources of language used to measure emotion, if 'self-report' can also be understood to include elicited natural language (i.e., free-text or open responses). This seems against the authors' best interests.

5. In their discussion, I would like to see the authors consider whether their results would extend across languages. All data collection and analysis were in English, a known outlier both culturally and linguistically (e.g., Blasi et al., 2022). Is there evidence suggesting the observed relationships would hold in other languages? What are the practical barriers to testing this, or for researchers who don't work with English data to implement the analyses described?

6. The lack of association between language measures and self-report in Datasets 3a and 3b is taken to mean the self-reports were not accurate. I'm not convinced of this interpretation. One alternative possibility is that it has to do with the register or function of the language – it occurred in conversation, which means that people may have been speaking differently than they would by themselves. Further, there is some evidence to suggest that language estimates of valence correlate with self-reported valence when the former are elicited (e.g., Carlier et al., 2021; Hoemann et al., 2024), but not when the language is unknowingly produced (e.g., Sun et al., 2020). I would like to see the authors elaborate on these considerations in their discussion.

7. In the analysis of facial movements, there is an imbalance between the number of positively-valenced emotions and the number of negative-valenced emotions. Is there a way to average across the likelihoods for negatively-valenced faces?

8. The authors found that language measures of valence showed significant associations with facial movements and vocal cues, but that language measures of discrete emotions did not. I'd like to see this tied more directly to the existing literature on facial and vocal cues. Does it sync with what we already know about whether these nonverbal behaviors can be uniquely assigned to emotion categories?

More minor points:

9. The title "Self-report, behavior, and emotion words..." should be changed, in my opinion; there is no focus on emotion words (i.e., labels for emotions) in this paper.

10. Can the authors give examples of 'emotion-related words' (page 5), to illustrate that these are not only words for emotion (e.g., "happy") but words with affective connotation, related to emotional events, etc. (e.g., "smile," "party," "good")?

11. LIWC measures the relative frequency of emotion word use, or the proportion of all words in a text that are related to emotion (or any other dictionary category) (page 6).

12. I'm not sure of the value of reporting the associations between language estimates of valence and self-report measures of discrete emotions in the main text. It seems that researchers interested in valence would ask participants to report just that, rather than to endorse specific emotions. I might consider moving these results to the supplement.

References

- Blasi, D. E., Henrich, J., Adamou, E., Kemmerer, D., & Majid, A. (2022). Over-reliance on English hinders cognitive science. *Trends in Cognitive Sciences*. <https://doi.org/10.1016/j.tics.2022.09.015>
- Carlier, C., Niemeijer, K., Mestdagh, M., Bauwens, M., Vanbrabant, P., Geurts, L., van Waterschoot, T., & Kuppens, P. (2021). In search of state and trait emotion markers in mobile-sensed language: A field study. *JMIR Mental Health*. <https://doi.org/10.2196/31724>
- Hoemann, K., Warfel, E., Mills, C., Allen, L., Kuppens, P., & Wormwood, J. B. (2024). Using freely generated labels instead of rating scales to assess emotion in everyday life. *Assessment*, 10731911241283623. <https://doi.org/10.1177/10731911241283623>

Reviewer #2 (Remarks to the Author):

This article addresses the use of language to measure emotion, but several key aspects require further clarification and refinement to improve the manuscript's clarity and contribution.

1. A primary issue lies in the distinction between language and self-report. Both constructs involve verbal responses, making it difficult for readers to discern how the authors distinguish between them. For instance, open-ended self-reports also utilize language to measure emotion. As such, language measures is a broad concept, and the article would benefit from clear definitions at the outset. I recommend that the authors provide precise definitions of language measures, self-reports, and other tools they use to assess emotions. Distinguishing these measures clearly in the introduction will improve conceptual clarity and guide the reader.

2. Another significant concern is whether the different measures (e.g., language, self-report, facial expressions, and vocal expressions) all capture responses to the same emotional event. From the article, it seems that the language-based measures reflect a retrospective description of positive or negative events, which introduces variability in the content of the emotional expression. This differs from self-reports, facial expressions, and vocal expressions, which are typically used to assess the protagonist's direct emotional response. If the underlying emotional content differs, it raises questions about why correlations between language measures and other emotional measures should be expected. Clarifying this distinction will help contextualize the findings and manage reader expectations.

3. The article mentions the use of facial and vocal expression analyses but lacks sufficient detail on the software and parameters used for these assessments. I encourage the authors to specify which software tools (e.g., Facereader, PRAAT) were used and outline the key parameters analyzed for each measure. Additionally, it would be valuable to discuss the validity and reliability of these tools to strengthen the credibility of the methodology.

4. While the introduction offers relevant background, several sections of the literature review appear incomplete or lack

proper organization. For example, the discussion around “using language when other measures are unavailable” and “using language to overcome limitations of other measures” contains overlapping content that could be consolidated. Integrating these sections would improve the flow of the argument and avoid redundancy.

5. In the section titled “What is the association between language operationalizations and measures often used to assess emotion,” the focus is primarily on the relationship between language measures and self-reports. However, the title suggests that other measures, such as facial or vocal expressions, would also be discussed. I recommend expanding this section to address how these additional measures are related to language measures. Moreover, it is important to explain how language and self-reports were operationalized and assessed in these experiments. Without this clarity, it becomes difficult to evaluate the specific challenges in previous studies and the ways in which the current research addresses them.

EDITORIAL POLICIES

We ask that you ensure your manuscript complies with our editorial policies and reporting requirements.

To that end, we require revised manuscripts to be accompanied by two completed items: a reporting summary that collects information on study design and procedure, and an editorial policy checklist that verifies compliance with all required editorial policies.

- <https://www.nature.com/documents/nr-reporting-summary.zip>>Nature Research Reporting Summary
- <https://www.nature.com/documents/nr-editorial-policy-checklist.pdf>>Editorial Policy Checklist

All points on the policy checklist must be addressed. Your revised manuscript can only be sent back to the referees if these checklists are completed and uploaded with the revision.

Notes: If you have submitted a Stage 1 Registered Report, Review, Primer, Comment, or Perspective you do not need to submit these forms. If you have already submitted these forms, you may disregard this request.

** Visit Nature Research's author and referees' website at <http://www.nature.com/authors>>www.nature.com/authors for information about policies, services and author benefits**

If you experience problems in linking your ORCID, please contact the <http://platformsupport.nature.com/>>Platform Support Helpdesk.

Version 1:

Decision Letter: second round

Dear Ms Munin,

Your manuscript titled "Self-report, Behavior, and Words: Examining Associations Between Language and Other Measures Used to Study Emotion" has now been seen by our reviewers, whose comments appear below. In light of their advice I am delighted to say that we are happy, in principle, to publish a suitably revised version in Communications Psychology.

We therefore invite you to revise your paper one last time to address the remaining concerns of our reviewers and a list of editorial requests. Please ensure you discuss the remaining reviewer concerns as limitations of the study. At the same time we ask that you edit your manuscript to comply with our format requirements and to maximise the accessibility and therefore the impact of your work.

EDITORIAL REQUESTS:

SUBMISSION INFORMATION:

OPEN ACCESS:

* DATA AVAILABILITY:

Link Redacted

Best regards,

Jennifer Bellingtier

Jennifer Bellingtier, PhD
Senior Editor
Communications Psychology

Yafeng Pan, PhD
Editorial Board Member
Communications Psychology
orcid.org/0000-0002-5633-8313

REVIEWERS' EXPERTISE:

Reviewer #1: Emotion and language
Reviewer #2: Emotional measures

REVIEWERS' COMMENTS:

Reviewer #1 (Remarks to the Author):

My thanks to the authors for their responsive and thorough revisions to this manuscript, in light of my and the other reviewers' comments. I am glad to see the sensitive use of language and the addition of methodological and theoretical considerations. This will be a useful reference for many!

Reviewer #2 (Remarks to the Author):

I would like to thank the authors for their thoughtful responses to my previous queries and for making careful revisions. Many of my concerns have been addressed; however, new questions have arisen, particularly regarding the language measures, self-reports, facial expressions, and vocal expressions.

Key Concerns:

Measurement of Emotion: While I appreciate the clarification provided, I still do not believe that the language measures, self-reports, facial expressions, and vocal expressions measure the same emotional content. These modalities may co-occur in emotional experiences, but this does not necessarily mean they reflect the same emotional content. For example, when a person is happy, they might be more likely to express this emotion through facial expressions rather than vocalizations in certain contexts. I believe the current explanation that all these measures tap into the same emotional content is problematic and needs further clarification. The relationship between these different expressions and emotional content should be addressed with greater precision.

Consistency in Coding: Another concern relates to the use of different automatic coding software across various databases. This variability in the software used to code facial expressions and vocal expressions could undermine the consistency of these measurements. To ensure greater reliability and validity in the data, I recommend that the authors use a single, standardized coding software across all datasets to improve measurement consistency.

Reviewer #1:

1. I recommend more precise wording for what is being assessed in language. Valence is not the same as emotion; it is one (critical) dimension of emotion. Valence does not capture the richness of emotional experience; language-estimated valence is also not perfectly correlated with self-report, even when the latter is maximally reliable. It seems more felicitous to refer to 'affect' as the broader construct being estimated in language, or even simply to refer to valence. This is especially so given that valence was the only self-/other-report measure collected in Datasets 2 and 3.

Thank you for highlighting how we can improve our description of the dimensions of emotion assessed by our language measures. We have revised the language in the Abstract, Introduction, and Discussion to clarify that the language measures provide estimates of valence (or discrete emotions), rather than estimates of emotion broadly. Similarly, we have clarified the wording in the manuscript to distinguish between using language measures to *study* emotion (i.e., address research questions relating to emotion) versus to *assess* or *estimate* properties of emotion such as valence and discrete emotions. Revisions made in relation to this point are indicated in bold below:

Abstract

(p. 2) “Furthermore, we examine associations across different dictionaries (LIWC-22, NRC, Lexical Suite, ANEW, VADER) used to **estimate valence (i.e., positive versus negative emotion) or discrete emotions (i.e., anger, fear, sadness)** in language.”

(p. 2) “Results did not tend to significantly vary across [“**emotion**” omitted] dictionaries.”

Introduction

(p. 3) “However, little empirical attention has been paid to understanding whether researchers who select language as their measure of **valence (i.e., positive and/or negative emotion) or discrete emotions (e.g., anger, fear, sadness)** can consider language to be a likely correlate of other measures used to assess emotion (e.g., Donnellan & Warren, 2022; Jones et al., 2016; Li, 2022).”

(p. 3) “The present study expands the scope of previous investigations by testing associations between language **measures of valence and discrete emotions** and other measures that are often used to assess emotion in three large, multimodal datasets.”

(pp. 4-5) “First, **researchers can often employ language measures [omitted: “language can help researchers measure emotion”]** in contexts where self-report or behavioral coding measures are absent. For example, a large body of work has used language methods to **assess valence and/or discrete emotions** in online spaces, such as social media platforms (Garcia & Rimé, 2019; Jones et al., 2019; Kaur et al., 2021), consumer review websites (Malik & Hussain, 2017; Rocklage et al., 2018; Rocklage &

Fazio, 2020), blogs (M.A. Cohn et al., 2004), and forums (Chen et al., 2020). Researchers also use language to **estimate valence and/or discrete emotions** in text sources such as written diaries (Pennebaker & King, 1999; Tov et al., 2013) and historical text corpora (Morin & Acerbi, 2017; Rheault et al., 2016). Moreover, researchers leveraging archival datasets that lack self-report or behavioral measures of emotion may use language to measure **valence and/or discrete emotions** (e.g., Li, 2022).”

(p. 5) “**Analysis of naturalistic language may therefore provide a promising alternative to self-report when assessing valence and/or discrete emotions over long timeframes.**”

(p. 6) “What Is the Association Between Language Operationalizations and Measures Often Used to **Study** Emotion?”

(p. 7) “The relations between language operationalizations and other measures often used to **study** emotion have been largely untested, and the few previous investigations are somewhat constrained in their scope.”

(p. 7) “For example, research in computational linguistics finds that language measures are associated with observer ratings of the writer’s **valence** in online written posts (e.g., Ribeiro et al., 2016; Schwartz et al., 2013).”

(p. 8) “To date, however, no previous work has tested whether language **measures of valence and/or discrete emotions** are associated with facial cues or vocal measures.”

(p. 8) “Therefore, research is needed to test associations between a wider range of language dictionaries and other measures often used to **study** emotion.”

(p. 8) “To better understand how language relates to other measures often used to **study** emotion, the present study tests associations between a broader range of [**“emotion” omitted**] dictionaries (i.e., LIWC-22, NRC, Lexical Suite, ANEW, VADER) and a broader range of measures (i.e., self-report, observer report, facial cues, and vocal cues), compared to previous work.”

Discussion

(pp. 20-21) “Researchers who want to study emotion but cannot use measures often used to assess emotion are increasingly turning to language (e.g., Donnellan & Warren, 2022; Jones et al., 2016; Li, 2022), yet limited previous work has tested associations between language and other [**“emotion” omitted**] measures. The present study examined 3 large, multimodal datasets to test whether language measures **of valence and discrete emotions** were associated with self-report, observer report, facial cues, and vocal measures.”

(p. 21) “Moreover, the selection of a particular [**“emotion word” omitted**] dictionary tended to not make a difference; patterns of associations across different dictionaries were highly correlated with one another.”

(p. 21) “The present findings suggest that the associations between language and other measures **used to study** emotion may depend on the other measures considered, and in some cases, the language measures involved.”

(p. 22) “Are [**“Emotion” omitted**] Language Dictionaries Less Susceptible to Display Rules?”

(p. 22) “A number of the [**“emotion” omitted**] dictionaries considered in the present study did not tend to show significant associations with facial cues but did show associations with other measures.”

(p. 27) “Associations did not tend to significantly vary across different dictionaries used by researchers to operationalize **valence or discrete emotions** in language.”

(p. 27) “Overall, the present work supports the idea that researchers who are unable to utilize self-report or behavioral coding may find language to be a useful measure [**“of emotion” omitted**] in their work.”

2. I disprefer the phrase ‘facial expression’ and the use of emotion category labels (e.g., ‘happy’) for them, when what’s really coded for is combinations of facial movements that are often associated with emotion categories. In the methods, the authors describe their facial coding algorithms as estimating the likelihood of particular facial expressions, which I don’t mind because it explicitly states that these are likelihoods. Overall, I find ‘cues’ (which is already in use) to be a more theory-neutral term.

We have revised the wording throughout all sections of the manuscript to replace the phrase “facial expression” with the phrase “facial cues”. We agree that the latter term better reflects the probabilistic nature of our coded facial measures, as well as the ongoing debate in the field about the diagnostic nature of facial movements for assessing emotion (e.g., Barrett et al., 2019; Keltner et al., 2019). See revisions below:

Abstract

(p. 2) “Building on previous work which focuses on associations between language and self-report, we test associations between language and a broader range of measures (self-report, observer report, facial **cues**, vocal cues).”

(p. 2) “Significant associations between language and facial **cues** emerged for language measures of valence but not for language measures of discrete emotions.”

(p. 2) “*Keywords:* language, emotion, dictionaries, self-report, facial **cues**, measurement”

Introduction

(p. 8) “To date, however, no previous work has tested whether language measures of valence and/or discrete emotions are associated with facial **cues** or vocal measures.”

(p. 8) “To better understand how language relates to other measures often used to assess emotion, the present study tests associations between a broader range of emotion dictionaries (i.e., LIWC-22, NRC, Lexical Suite, ANEW, VADER) and a broader range of measures (i.e., self-report, observer report, facial **cues**, and vocal cues), compared to previous work.”

Table 1

(p. 9) “Facial **Cues**”

Method

(p. 14) “Facial **Cues**”

(p. 14) “Facial **cues** were measured in Datasets 2 and 3.”

(p. 15) “Second, moment-to-moment self-report, observer-report, facial **cues** [**“expression” omitted**], and vocal measures in Datasets 2 and 3 were averaged across the whole video to test for between-person, rather than within-person, associations with language.”

Results

(p. 17) “Language measures varied in their associations with facial **cues** in two datasets”

(p. 17) “Language measures of valence significantly related to facial **cues**, but language measures of discrete emotions generally did not.”

(p. 17) “For individual correlations with angry, fearful, and sad facial **cues**, see Supplementary Table S4c.”

(p. 17) “However, language measures of discrete emotions did not tend to show significant associations with facial **cues**.”

Figure 1

(p. 20) “Forest plot of selected Spearman correlations between language measures of valence and self-report, observer report, facial **cues**, and vocal cues”

(p. 20) “Happy Facial **Cues**”

Discussion

(p. 21) “The present study examined 3 large, multimodal datasets to test whether language measures of valence and discrete emotions were associated with self-report, observer report, facial **cues**, and vocal measures. Language measures showed largely consistent patterns of significant associations with self-report and observer report. Language measures of valence also showed significant associations with facial **cues**, but language measures of discrete emotions did not.”

(p. 22) “However, language less consistently related to facial [**“expressions” omitted**] and vocal cues. When facial **cues** were considered, language measures of valence showed patterns of significant associations, yet language measures of discrete emotions did not.”

(p. 22) “The present work also raises the possibility that associations between language and facial **cues** may be stronger for dictionaries which operationalize valence compared to dictionaries which operationalize discrete emotions.”

(pp. 22-23) “A number of the dictionaries considered in the present study did not tend to show significant associations with facial **cues** but did show associations with other measures. In addition to the perspectives on variable associations between discrete emotions and facial **cues** discussed above, another possible explanation for the inconsistent associations with facial **cues** may be the operation of display rules (Ekman, 1993). That is, there may have been reduced association due to norms against facial displays of negative feelings, whereas negative feelings may have seeped into language relatively less unchecked. Consistent with the possibility that suppression of negative facial **displays** may underlie the lack of association with language, previous work suggests that correlations between measures of emotion diminish when facial **movements** are suppressed (Dan-Glauser & Gross, 2013). Indeed, in Dataset 2, facial **cues** of anger, fear, and sadness tended to be weaker predictors of self-reports of emotion than language measures were (see Supplementary Table S3b), suggesting that facial **cues** may have less reliably tracked with negative feelings. In Datasets 3a and 3b, facial **cues** tended not to significantly relate to self-report in the expected direction, though it is unclear whether the lack of association may stem from issues with the self-report measures (see below).”

(p. 24) “For example, a greater number of significant associations emerged between language measures and both facial **cues** and vocal pitch in dyadic conversations (Datasets 3a and 3b) than in spoken monologues to a camera (Dataset 2).”

(p. 25) “Furthermore, the rapid advancement of machine learning algorithms to extract emotion measures from video data (S. Li & Deng, 2022) raises the possibility that the algorithms which quantified the facial [**“expressions” omitted**] and vocal measures in the monologue task may have been less precise than the more recently developed models which quantified these measures in the conversation task.”

3. There are several recent papers that the authors could incorporate into their literature review and discussion. For example, Carlier et al (2021) investigated the relationship between self-

report and estimates of valence and emotion across multiple modalities (e.g., spoken natural language, vocal acoustics, text messages), finding weak correlations overall and modest results for valence and happiness. Hoemann et al (2024) found that labels for current experience recapitulated valence ratings.

Thank you for bringing these relevant articles to our attention. We have incorporated the papers by Carlier et al (2022) and Hoemann et al (2024) into our Introduction and Discussion. In response to a comment by Reviewer #2, we have also expanded our literature review in the Introduction to describe these and other previous studies which assess relations between language and other measures in greater depth. See revisions below:

Introduction

(p. 7) **“Other work found significant correlations between language and self-reports of valence and discrete emotions (i.e., anger, anxiety, sadness) for prompted spoken language, but not for naturalistic written language (i.e., text messages) (Carlier et al., 2022).”**

(p. 7) **“And although one recent study found significant associations between self-reports of valence and estimates of valence from participants’ written descriptions of their emotions (Hoemann et al., 2024), other studies did not find any significant associations between self-reports of valence and estimates of valence from either naturalistic spoken language (Sun et al., 2020) or social media posts (Kross et al., 2019).”**

(p. 8) **“Furthermore, in psychology publications, most previous work has focused on LIWC (e.g., Kross et al., 2019; Sun et al., 2020; but see Hoemann et al., 2024), and less is known about the associations of other dictionary measures (e.g., word weighting and rule-based approaches).”**

Discussion

(p. 21) **“These findings align with some previous studies which found significant correlations between language measures and self-reports (e.g., Carlier et al., 2022; Hoemann et al., 2024; Tov et al., 2013) and observer reports (Liess et al., 2008).”**

(p. 24) **“For example, language does not tend to significantly relate to self-report when sampled from text messages, social media posts, or everyday spoken language (Carlier et al., 2022; Kross et al., 2019; Sun et al., 2020). Yet language has been found to significantly relate to self-report when participants are prompted to write or speak about an emotional experience (Carlier et al., 2022; Hoemann et al., 2024; Kahn et al., 2007).”**

4. I wasn't sure how to feel about the argument that repeated real-time self-reports may impact the emotional process (page 5). I do not know what the 'emotional process' is, or how we can get away without impacting it whenever we explicitly ask participants to reflect and respond in ways

and at times they might not have otherwise. More critically, this argument seems to constrain the types or sources of language used to measure emotion, if 'self-report' can also be understood to include elicited natural language (i.e., free-text or open responses). This seems against the authors' best interests.

You raise some important points for our argument about repeated self-report and how these claims relate to our focus on language. We have revised the wording in this section to be more specific about the aspect of the “emotional process” impacted in previous research on repeated self-reports (see below). We also wish to clarify that our stance is not that any single real-time measurement will influence the emotional response, but rather that long timeframes (e.g., 30 minutes) present measurement constraints that may limit the utility of self-report. If researchers ask participants to provide a self-report immediately following a long task, retrospective biases may color the emotions participants report experiencing across the task (e.g., Fredrickson, 2000). An alternative approach, then, could be to repeatedly probe participants' emotion throughout the task, but this approach is also not without its problems as it too may alter participants' ratings (Johar & Sackett, 2018). Thus, spontaneously generated, naturalistic language (e.g., a conversation transcript) could provide an unobtrusive measure of emotion across the duration of a long task, overcoming these potential limitations of self-report. Indeed, previous work has also argued that unobtrusive measurement over long periods of time is a benefit of using language, rather than repeated self-report, to study emotion (Carlier et al., 2022, p. 2).

Introduction

(p. 5) “For instance, self-reporting emotion may be challenging over a long timeframe. Participants may overemphasize or underemphasize emotions which fluctuate across a long time when asked for a single retrospective report of their emotion (Fredrickson, 2000; Robinson & Clore, 2002) or another person's emotion (Goldenberg et al., 2022). **Yet repeated 'real time' self-report measurements aiming to track emotional fluctuations over a long duration may dampen participants' emotion ratings (Johar & Sackett, 2018). Analysis of naturalistic language may therefore provide a promising alternative to self-report when assessing valence and/or discrete emotions over long timeframes.**”

The revised manuscript now mentions that this example of how language can overcome limitations of self-report over longer timeframes is specific to approaches that focus on naturalistic language, rather than the prompted language you describe. As noted below in our response to Point #1 from Reviewer #2, we consider the quantitative analysis of free-response text to fall under the category of language measures rather than self-report, and we have provided definitions in the revised manuscript to make this distinction more clear (see revision below). We believe that our argument does not constrain the use of prompted language from free-response items, as prompted language may be used to overcome other perceived limitations of self-report and behavioral coding described in this section of the Introduction (e.g., automated text analysis of prompted language is less time- and resource-intensive than manual coding approaches).

Introduction

(p. 4) **“In contrast to these other approaches, language methods can be defined as involving the quantitative analysis of natural language, such as the words generated in response to an open-ended prompt (i.e., prompted language) or the words produced in everyday speech or writing (i.e., naturalistic language) (Mohammad, 2021; Tausczik & Pennebaker, 2010).”**

5. In their discussion, I would like to see the authors consider whether their results would extend across languages. All data collection and analysis were in English, a known outlier both culturally and linguistically (e.g., Blasi et al., 2022). Is there evidence suggesting the observed relationships would hold in other languages? What are the practical barriers to testing this, or for researchers who don't work with English data to implement the analyses described?

The revised Discussion now includes a “Limitations” section in which we discuss (among other methodological limitations) the generalizability of our results given that our research was conducted in English. We acknowledge that English is an outlier among world languages (Blasi et al., 2022), though previous research on language and emotion often focuses on English speakers (Mohammad, 2021). We speculate that our results might generalize beyond English, as some previous evidence suggests that correlations among measures of emotion (e.g., self-report, verbal responses, nonverbal behavior) may be consistent across English and non-English-speaking cultures (Matsumoto et al., 2007). Moreover, prior work suggests that estimates of positive valence from text remain stable when the words are translated into other languages (Dodds et al., 2015). Ultimately, future research is necessary to test the prediction that our findings are generalizable to other language contexts. The revised manuscript notes that two dictionaries considered in the present study (LIWC and NRC) have already been translated into several languages. We hope that this information can help guide readers who may be interested in testing replication with non-English data.

Discussion

(pp. 25-26) **“First, all analyses in the present study were conducted with American English-speaking samples. English is known to be a linguistic and cultural outlier among world languages (Blasi et al., 2022), and most prior work on language and emotion only considers English speakers (see Mohammad, 2021). Still, it is possible that associations between language and other measures used to study emotion may generalize to other languages beyond English. For example, ratings of positive valence for words in one language tend to remain stable when the words are translated into other languages (Dodds et al., 2015). Additionally, some previous work finds that correlations among measures of emotion vary little between English-speaking and non-English-speaking cultures (Matsumoto et al., 2007). Future research with non-English-speaking samples is ultimately needed to test whether the present results indeed replicate across languages. Of the dictionaries tested in the present study, LIWC has been translated into at least 14 languages (Boyd et al., 2022), and the NRC Emotion Lexicon has been translated into over 100 languages (Mohammad, 2021), which can facilitate tests in non-English contexts.”**

6. *The lack of association between language measures and self-report in Datasets 3a and 3b is taken to mean the self-reports were not accurate. I'm not convinced of this interpretation. One alternative possibility is that it has to do with the register or function of the language – it occurred in conversation, which means that people may have been speaking differently than they would by themselves. Further, there is some evidence to suggest that language estimates of valence correlate with self-reported valence when the former are elicited (e.g., Carlier et al., 2021; Hoemann et al., 2024), but not when the language is unknowingly produced (e.g., Sun et al., 2020). I would like to see the authors elaborate on these considerations in their discussion.*

Thank you for raising an alternative interpretation of our findings. We have revised the Discussion section to highlight the possibility that the distinction between language in naturalistic contexts (e.g., a “getting-to-know-you” conversation) and language in contexts where participants are prompted to discuss their emotional experience (e.g., a monologue about an emotional event) may underlie the inconsistent associations we observed. Furthermore, we have revised our language to soften our position on whether timing shapes the relative usefulness of language and self-report in light of this alternative interpretation. See revisions in bold below:

Discussion

(p. 23) **“Does Timing Shape When Language Rather Than Self-Report May Be Most Useful?”**

(p. 23) **“Why did language measures not significantly relate to self-reported emotion in a 30-minute conversation (Datasets 3a and 3b) but did significantly relate to self-report in a 2-minute monologue task (Dataset 2)? One possibility is that the current findings lend credence to concerns about the limitations of self-report when measuring emotion over long periods of time and support the preference for language measures under those circumstances.”**

(p. 24) **“However, another interpretation is that the present findings are consistent with some prior work suggesting that self-reported emotion may relate less strongly to language drawn from naturalistic settings than language generated in response to an experimental prompt. For example, language does not tend to significantly relate to self-report when sampled from text messages, social media posts, or everyday spoken language (Carlier et al., 2022; Kross et al., 2019; Sun et al., 2020). Yet language has been found to significantly relate to self-report when participants are prompted to write or speak about an emotional experience (Carlier et al., 2022; Hoemann et al., 2024; Kahn et al., 2007). In the present study, the monologue about an emotional event (Dataset 2) more closely resembles the prompted context, whereas the “getting-to-know-you” conversation (Datasets 3a and 3b) may be more similar to a naturalistic setting. Thus, future research may wish to more systematically test whether naturalistic or prompted contexts may influence associations between language and self-report.”**

(p. 27) “The findings **may also be considered** consistent with concerns that self-report may inadequately measure emotions over long timescales and raise the possibility that language measures may be preferable in such circumstances.”

7. In the analysis of facial movements, there is an imbalance between the number of positively-valenced emotions and the number of negative-valenced emotions. Is there a way to average across the likelihoods for negatively-valenced faces?

Thank you for your suggestion. The revised manuscript now reports correlations between language measures of valence and likelihoods of “negative faces”, which are defined as the average of angry, fearful, and sad facial cues. The results are largely consistent with the previously reported correlations between language and angry facial cues, with the exception of 1 additional significant correlation in Dataset 2 for negative faces which was not significant for angry faces (all results reported post-Bonferroni correction). For completeness, we report the individual correlations between language measures of valence and angry, fearful, and sad faces in the Supplemental material. See revisions below:

Results

(p. 17) “Additionally, **2** language measures of valence **were** significantly associated with lower likelihoods of **negative** faces (**i.e., average of angry, fearful, and sad facial cues**) in Dataset 2 ($\rho = -.28$ to $.01$), and **3** language measures of valence were significantly associated with lower likelihoods of **negative** faces in both Datasets 3a ($\rho = -.25$ to $-.05$) and 3b ($\rho = -.27$ to $-.06$). For **individual** correlations with **angry, fearful, and sad** facial cues, see Supplementary Table S4c.”

8. The authors found that language measures of valence showed significant associations with facial movements and vocal cues, but that language measures of discrete emotions did not. I'd like to see this tied more directly to the existing literature on facial and vocal cues. Does it sync with what we already know about whether these nonverbal behaviors can be uniquely assigned to emotion categories?

We have revised the Discussion to more directly refer to previous literature which speaks to the extent to which facial and vocal cues map onto measures of valence and/or discrete emotions. The revised manuscript now raises the possibility that the lack of significant associations between language measures of discrete emotions and facial cues aligns with perspectives which suggest that patterns of facial cues may not be uniquely associated with discrete emotion categories. Furthermore, we raise the possibility that the inconsistent associations between language measures and vocal cues could reflect weaker ties between vocal cues and valence relative to other affective properties (i.e., arousal). See revisions below:

Discussion

(p. 22) “**The present findings may contribute to ongoing discussion about the extent to which nonverbal cues relate to measures of valence or discrete emotions. That is, some researchers argue that facial cues do not reliably map onto measures of**

discrete emotions (e.g., Barrett et al., 2019; but see also Keltner et al., 2019). Moreover, while previous research suggests that vocal pitch and vocal intensity are associated with discrete emotions (e.g., Sauter et al., 2010), vocal cues may be more strongly related to physiological arousal than valence (see Scherer et al., 2002 for a review). Inconsistent associations between nonverbal cues and language measures of valence and discrete emotions may be considered consistent with these views.”

(p. 22) “In addition to the perspectives on variable associations between discrete emotions and facial cues discussed above, another possible explanation for the inconsistent associations with facial cues may be the operation of display rules (Ekman, 1993).”

More minor points:

9. The title “Self-report, behavior, and emotion words...” should be changed, in my opinion; there is no focus on emotion words (i.e., labels for emotions) in this paper.

We have revised the title to better reflect the broad focus on words in our manuscript (e.g., emotion words and sentiment words, see point #10 below), rather than the narrow focus on labels for emotions.

Title Page

(p. 1) “Self-report, Behavior, and [**“Emotion” omitted**] Words: Examining Associations Between Language and Other Measures Used to Study Emotion”

Introduction

(p. 3) “Self-report, Behavior, and [**“Emotion” omitted**] Words: Examining Associations Between Language and Other Measures Used to Study Emotion”

10. Can the authors give examples of ‘emotion-related words’ (page 5), to illustrate that these are not only words for emotion (e.g., “happy”) but words with affective connotation, related to emotional events, etc. (e.g., “smile,” “party,” “good”)?

Thank you for this suggestion to improve the clarity of the manuscript, particularly for readers who are unfamiliar with the words included in these dictionaries. We have revised the paragraph on page 5 accordingly:

Introduction

(p. 5) “A common approach is to use pre-made word lists called dictionaries to identify words within a given text **that directly (e.g., “happy”, “sadness”) and/or indirectly (e.g., “good”, “funeral”) relate to emotion** (Boyd & Schwartz, 2021; Rocklage & Rucker, 2019).”

11. *LIWC measures the relative frequency of emotion word use, or the proportion of all words in a text that are related to emotion (or any other dictionary category) (page 6).*

Thank you for highlighting an opportunity for us to improve our wording in the manuscript. We have revised the passage you identified to reflect that LIWC measures the relative frequency, rather than the absolute frequency, of dictionary words in a given text.

Introduction

(p. 6) “Word counting dictionaries such as Linguistic Inquiry and Word Count (LIWC: Pennebaker et al., 2022) and the NRC Emotion Lexicon (Mohammad & Turney, 2010, 2013) measure the **relative** frequency of emotion-related word use (see Boyd & Schwartz, 2021).”

12. *I'm not sure of the value of reporting the associations between language estimates of valence and self-report measures of discrete emotions in the main text. It seems that researchers interested in valence would ask participants to report just that, rather than to endorse specific emotions. I might consider moving these results to the supplement.*

We wish to clarify that we report associations between language measures of valence and the self-report valence composite (i.e., average positive minus average negative ratings) derived from the self-report measures of discrete emotions in Dataset 1, rather than associations between language measures of valence and the self-report measures of discrete emotions themselves. Our approach is consistent with previous literature which examines the associations between language measures of valence and positive and/or negative composite ratings derived from self-reports of discrete emotions (e.g., the PANAS) (Kahn et al., 2007; Tov et al., 2013). Furthermore, some previous research has also investigated associations between language measures of valence (e.g., LIWC negative emotion) and self-report measures of discrete emotions (e.g., sadness: Carlier et al., 2022; Kahn et al., 2007; Tov et al., 2013). Therefore, while we recognize that a self-report measure of valence would provide a more direct comparison to language measures of valence, we feel that the analyses reported in the manuscript are relevant to researchers who use language to study emotion.

The revised manuscript notes how some previous research has relied on composite measures of valence derived from discrete emotion scales such as the PANAS when testing associations between language measures and self-reports of valence:

Introduction

(p. 7) “**For example, one study found significant associations between spoken language following a film task and self-ratings of emotion (i.e., the PANAS: Watson et al., 1988) for positive valence and amusement, but not for negative valence and sadness (Kahn et al., 2007).**”

Reviewer #2:

1. A primary issue lies in the distinction between language and self-report. Both constructs involve verbal responses, making it difficult for readers to discern how the authors distinguish between them. For instance, open-ended self-reports also utilize language to measure emotion. As such, language measures is a broad concept, and the article would benefit from clear definitions at the outset. I recommend that the authors provide precise definitions of language measures, self-reports, and other tools they use to assess emotions. Distinguishing these measures clearly in the introduction will improve conceptual clarity and guide the reader.

Thank you for raising the importance of defining the measures we assess in our study. In the revised Introduction, we have added our definitions of self-report, behavioral coding, and language approaches. For instance, we define self-reports of emotion as closed-response questionnaires about participants' subjective feelings. By contrast, we define language measures as measures which quantify psychological constructs from written and/or spoken language (e.g., the dictionary approaches described in the Introduction). We define language broadly, as previous research investigating associations between language measures and self-reports of emotion has considered language from a range of contexts, including both open-ended prompts (e.g., Hoemann et al., 2024) and language drawn from everyday life (e.g., Kross et al., 2019).

Introduction

(p. 4) **“Researchers have several measurement options available for studying emotion. For example, a long history of research in emotion has used closed-ended self-report scales to assess participants’ subjective feelings (e.g., Watson et al., 1988; see also Diener et al., 2018). Other work relies on automated or manual coding of nonverbal behaviors, such as facial or vocal cues (e.g., J. F. Cohn et al., 2007; Eyben et al., 2016). In contrast to these other approaches, language methods can be defined as involving the quantitative analysis of natural language, such as the words generated in response to an open-ended prompt (i.e., prompted language) or the words produced in everyday speech or writing (i.e., naturalistic language) (Mohammad, 2021; Tausczik & Pennebaker, 2010).”**

2. Another significant concern is whether the different measures (e.g., language, self-report, facial expressions, and vocal expressions) all capture responses to the same emotional event. From the article, it seems that the language-based measures reflect a retrospective description of positive or negative events, which introduces variability in the content of the emotional expression. This differs from self-reports, facial expressions, and vocal expressions, which are typically used to assess the protagonist's direct emotional response. If the underlying emotional content differs, it raises questions about why correlations between language measures and other emotional measures should be expected. Clarifying this distinction will help contextualize the findings and manage reader expectations.

Researchers often consider language to reflect attention (Boyd & Schwartz, 2021; Tausczik & Pennebaker, 2010)—that is, even if participants are writing (e.g., Dataset 1) or talking (e.g., Dataset 2) about a past emotional event, their word choice provides information about their present attentional focus (and by extension, their current emotional state). Therefore,

we consider the language measures in the present study to each assess participants' direct emotional response. Indeed, in Dataset 2, participants generated their language concurrently with their facial and vocal cues. Moreover, in Datasets 3a and 3b, the language measures are derived from conversation transcripts, which are tied to participants' present psychological state.

In most of the correlations tested in the present work, language is considered in relation to another "real-time" measure of emotion (e.g., facial cues). There are two exceptions, both involving associations between language measures and retrospective self-reports. In Dataset 1, language measures from a narrative retelling of a social rejection event significantly related to participants' retrospective self-reports of discrete emotions experienced during the event. In Datasets 3a and 3b, language measures from conversation transcripts did not significantly relate to retrospective self-reports of participants' valence during the conversation (nor did language significantly relate to momentary pre-conversation and post-conversation valence self-reports). We also did not tend to observe significant correlations in the expected direction between self-report and other real-time measures (e.g., facial cues) in Datasets 3a and 3b.

That said, previous research suggests that although we can expect measures which assess direct emotional reactions to correlate most strongly, correlations between direct measures and retrospective accounts of emotion may still emerge. For example, Tov and colleagues (2013) found that language measures of negative emotion (i.e., LIWC) from written reports of daily events significantly correlated with both daily self-reports of negative emotion as well as with retrospective self-reports of emotion over the sampling period (3 weeks or 2 months) in two studies. Therefore, it is reasonable to expect that correlations between momentary and retrospective measures of emotion could emerge the present study, especially given that the timescales of each dataset (i.e., 2–30 minutes) were considerably shorter than in Tov et al (2013).

In the revised Introduction, we orient readers to the use of both prompted language and naturalistic language as measures of participants' direct emotional response. Furthermore, the revised Method section clarifies the description of our self-report measures to assist readers with understanding whether each measure reflects a direct emotional response or a retrospective account of the emotional event:

Introduction

(p. 4) **“In contrast to these other approaches, language methods can be defined as involving the quantitative analysis of natural language, such as the words generated in response to an open-ended prompt (i.e., prompted language) or the words produced in everyday speech or writing (i.e., naturalistic language) (Mohammad, 2021; Tausczik & Pennebaker, 2010). As researchers often consider language to reflect attentional focus (Boyd & Schwartz, 2021), language measures derived from both prompted language and naturalistic language have been used to assess participants' current affective state (e.g., Jones et al., 2016; Kahn et al., 2007).”**

Method

(p. 13) “Self-reports of emotion were measured in all three datasets. In Dataset 1, participants responded to 16 items about discrete emotions they **remembered experiencing** during the recalled rejection on a 1 (*not at all*) to 5 (*extremely*) scale.”

(p. 13) “In Dataset 3, participants provided self-reports of their emotion at three timepoints: **a momentary account of their emotion** before the conversation, **a momentary account of their emotion** after the conversation, and a retrospective account of how they **had** felt during the conversation (**reported immediately after the conversation**).”

3. The article mentions the use of facial and vocal expression analyses but lacks sufficient detail on the software and parameters used for these assessments. I encourage the authors to specify which software tools (e.g., Facereader, PRAAT) were used and outline the key parameters analyzed for each measure. Additionally, it would be valuable to discuss the validity and reliability of these tools to strengthen the credibility of the methodology.

We have revised the Method section to include greater detail on the software programs and parameters which derived the facial and vocal cue measures used in Datasets 2 and 3. The extractions of facial and vocal features were conducted by the authors of the datasets, and these measures are included in each dataset. Therefore, we refer readers to the original papers for each dataset for further detail on the procedure. Revisions made in relation to this point are indicated in bold below:

Method

(p. 14) “In Dataset 2, an automated software program (**Emotient by iMotions**) extracted moment-to-moment measures of the likelihood of different facial expressions (e.g., happy, angry, fearful, sad) for the speaker in each video (**the original paper additionally used the software to extract the presence of 20 Facial Action Units; see Ong et al., 2021 for more detail**). In Dataset 3, **a convolutional neural network trained on a facial image database** extracted moment-to-moment measures of the likelihood of different facial expressions (e.g., happy, angry, fearful, sad) for each speaker in each conversation (**see Reece et al., 2023 for more detail on the feature extraction procedure**).”

(p. 14) “Vocal cues were measured in Datasets 2 and 3. In Dataset 2, an automated software program (**openSMILE v2.3.0**) extracted moment-to-moment extended **GeMAPS (eGeMAPS) parameter measures** of vocal pitch (i.e., fundamental frequency, F0) and vocal intensity (i.e., loudness) for the speaker in each video (**see Ong et al., 2021 for more detail on the feature extraction procedure**). In Dataset 3, an automated **software package (Parselmouth)** extracted moment-to-moment measures of vocal pitch (i.e., F0) **for each speaker in each conversation. Additionally, a model trained on an emotional speech dataset estimated moment-to-moment measures of vocal emotional intensity** for each speaker in each conversation (**see Reece et al., 2023 for more detail on the feature extraction procedure**).”

Furthermore, we have added discussion of the benefits and drawbacks of automated approaches to measuring facial and vocal cues in our revised Discussion section. In particular, we note that while automated approaches can show high accuracy when tested on controlled lab-based stimuli, performance drops have been observed when approaches are generalized to other contexts, such as real-world stimuli (Canedo & Neves, 2019). Thus, future research may benefit from additional measures of behavioral cues (e.g., human coding) to complement the approaches used in the present study.

Discussion

(p. 26) **“Second, the present study relied on automated machine learning approaches to measure facial and vocal cues, which may not always identify cues with high accuracy. For example, research suggests that automated approaches to measuring facial cues can perform well under controlled conditions yet show significantly greater error when applied to real-world stimuli (Canedo & Neves, 2019) or even when generalized to other lab-based datasets (Suresh et al., 2023). Moreover, the model used to identify facial cues in the present conversation dataset (Dataset 3) was not adapted to naturalistic conversation contexts, where facial movement tends to differ considerably from the prototypical poses used in training datasets (Reece et al., 2023). Therefore, future research with naturalistic data may wish to complement machine learning approaches with human-coded behavior to provide a more complete picture of facial and vocal cues which may relate to language approaches to studying emotion.”**

4. While the introduction offers relevant background, several sections of the literature review appear incomplete or lack proper organization. For example, the discussion around “using language when other measures are unavailable” and “using language to overcome limitations of other measures” contains overlapping content that could be consolidated. Integrating these sections would improve the flow of the argument and avoid redundancy.

Thank you for pointing out places where we can be more concise. We have revised the section you identified to consolidate our points and have also removed associated sub-headers. We feel that this revision has reduced redundancy and contributed to the readability of the manuscript. Revisions made in relation to this point are indicated in bold below:

Introduction

(pp. 4-5) **“[Sentence omitted] Compared to self-report and behavioral coding,** language offers at least two benefits for researchers interested in studying emotion. First, researchers can often employ language measures in contexts where self-report or behavioral coding measures are absent. **[Sentence omitted]** For example, a large body of work has used language methods to assess valence and/or discrete emotions in online spaces, such as social media platforms (Garcia & Rimé, 2019; Jones et al., 2019; Kaur et al., 2021), consumer review websites (Malik & Hussain, 2017; Rocklage et al., 2018; Rocklage & Fazio, 2020), blogs (M.A. Cohn et al., 2004), and forums (Chen et al., 2020). Researchers also use language to estimate valence and/or discrete emotions in text

sources such as written diaries (Pennebaker & King, 1999; Tov et al., 2013) and historical text corpora (Morin & Acerbi, 2017; Rheault et al., 2016). Moreover, researchers leveraging archival datasets that lack self-report or behavioral measures of valence and/or discrete emotions may use language measures instead (e.g., Li, 2022).”

(p. 5) **“Second, language measures tend to be unintrusive and easily implemented (Rocklage & Rucker, 2019) and therefore hold the potential to circumvent limitations of other measures. For instance, self-reporting emotion may be challenging over a long timeframe. Participants may overemphasize or underemphasize emotions which fluctuate across a long time when asked for a single retrospective report of their emotion (Fredrickson, 2000; Robinson & Clore, 2002) or another person’s emotion (Goldenberg et al., 2022). Yet repeated ‘real time’ self-report measurements aiming to track emotional fluctuations over a long duration may dampen participants’ emotion ratings (Johar & Sackett, 2018). Analysis of naturalistic language may therefore provide a promising alternative to self-report when assessing valence and/or discrete emotions over long timeframes. Furthermore, some researchers may lack access to resources to implement self-report or behavioral coding in their work. Collecting self-report or behavioral coding may require participant funds and/or time, lab space, or large teams of research assistants that may not be available. By contrast, research using language can be conducted with little cost beyond a computer and Wi-Fi access using public datasets and/or text scraped from the internet. Thus, automated language analysis can bypass self-report or resource intensive approaches like behavioral coding (Rocklage & Rucker, 2019).”**

5. In the section titled “What is the association between language operationalizations and measures often used to assess emotion,” the focus is primarily on the relationship between language measures and self-reports. However, the title suggests that other measures, such as facial or vocal expressions, would also be discussed. I recommend expanding this section to address how these additional measures are related to language measures. Moreover, it is important to explain how language and self-reports were operationalized and assessed in these experiments. Without this clarity, it becomes difficult to evaluate the specific challenges in previous studies and the ways in which the current research addresses them.

The revised manuscript expands the section on the relationship between language and other emotion measures in the ways you have suggested, and we feel that this revision has greatly strengthened our literature review. First, we describe previous studies in significantly greater detail with regard to the language and self-report measures used and highlight the specific findings from each. Second, we have added a paragraph break to draw readers’ attention to the section which describes previous research on associations with measures beyond self-report. Third, although no previous studies to our knowledge have tested associations between language and facial or vocal cues of emotion, we have expanded our background on the few studies which examined associations with observer reports (i.e., Liess et al., 2008; Sun et al., 2020).

Introduction

(p. 7) “The relations between language operationalizations and other measures often used to study emotion have been largely untested, and the few previous investigations are somewhat constrained in their scope. Some previous research has investigated the associations between language and self-reports of emotion and yielded a mix of significant and non-significant findings. **For example, one study found significant associations between spoken language following a film task and self-ratings of emotion (i.e., the PANAS: Watson et al., 1988) for positive valence and amusement, but not for negative valence and sadness (Kahn et al., 2007). Other work found significant correlations between language and self-reports of valence and discrete emotions (i.e., anger, anxiety, sadness) for prompted spoken language, but not for naturalistic written language (i.e., text messages) (Carlier et al., 2022). Another study found associations between written diary language and self-ratings of positive valence for daily assessments, but not for weekly assessments (Tov et al., 2013). And although one recent study found significant associations between self-reports of valence and estimates of valence from participants’ written descriptions of their emotions (Hoemann et al., 2024), other studies did not find any significant associations between self-reports of valence and estimates of valence from either naturalistic spoken language (Sun et al., 2020) or social media posts (Kross et al., 2019).**”

(pp. 7-8) “Less work has examined associations between language and measures other than self-reported emotion. For example, research in computational linguistics finds that language measures are associated with observer ratings of the writer’s emotion in online written posts (e.g., Ribeiro et al., 2016; Schwartz et al., 2013). Yet **very few studies have** tested associations with observer reports in offline contexts where other signals beyond language are available. **For example, one small study (N = 20) found significant associations between language and observational coding for negative emotion, but not for positive emotion (Liess et al., 2008), and another study did not find consistent associations between language and listeners’ valence ratings of audio clips (Sun et al., 2020). To date, however, no** previous work **has** tested whether language **measures of valence and/or discrete emotions are** associated with facial **cues** or vocal measures. Furthermore, in psychology publications, most previous work has focused on LIWC (e.g., Kross et al., 2019; Sun et al., 2020; **but see Hoemann et al., 2024**), and less is known about the associations of other dictionary measures (e.g., word weighting and rule-based approaches). Therefore, research is needed to test associations between a wider range of language dictionaries and other measures often used to assess emotion.”

THE UNIVERSITY OF TEXAS AT AUSTIN

January 31st, 2025

Dear Dr. Bellingtier and Dr. Pan,

Thank you and the two reviewers for your feedback on our revised manuscript “Language Measures Correlate With Other Measures Used to Study Emotion”. We are very pleased to learn that you plan to accept the manuscript for publication in *Communications Psychology*! We have revised the manuscript to comply with the journal’s formatting requirements and other points raised in the Editorial Requests Table (see table attached).

Below, we detail our responses to the additional points raised by Reviewer #2, as well as how we have addressed each point in the study limitations:

Measurement of Emotion: While I appreciate the clarification provided, I still do not believe that the language measures, self-reports, facial expressions, and vocal expressions measure the same emotional content. These modalities may co-occur in emotional experiences, but this does not necessarily mean they reflect the same emotional content. For example, when a person is happy, they might be more likely to express this emotion through facial expressions rather than vocalizations in certain contexts. I believe the current explanation that all these measures tap into the same emotional content is problematic and needs further clarification. The relationship between these different expressions and emotional content should be addressed with greater precision.

The reviewer raises an interesting point about whether language, self-report, facial cues, and/or vocal cues can be presumed to reflect the same emotional content. Some theories of emotion posit that emotional responding occurs synchronously across modalities, and therefore measures of emotion can be expected to correlate with one another (e.g., Ekman, 1992; Levenson, 1999). Previous literature on coherence across measures of emotion also finds support for associations across measures of emotion, though these associations tend to be imperfect (e.g., Mauss et al., 2005; Dan-Glauser & Gross, 2013).

That said, an exploration of whether measures truly reflect emotional content, or reflect emotional content to the same extent, is beyond the scope of the present study. In the revised limitations subsection, we discuss how the present work was not designed to test this issue, as our work instead focuses on whether researchers may consider language as an alternative to commonly used measures in emotion research. We briefly reflect on the possibilities for language, self-report, and behavior to tap into emotional content and/or sources other than one’s current emotional state and suggest that future research more directly examine this issue.

Discussion

(p. 21) **“Second, the present study tested whether different measurement approaches to emotion relate to one another but cannot speak to the degree to which these**

measures truly reflect emotional content. Several theories of emotion argue that measures of emotion should relate to one another because they reflect components of an emotional response^{65,66}. Still, it is possible that these measures could also tap into other processes. For example, emotion-related language could be used for purposes other than expressing one's current emotion (e.g., "she looks happy")⁴⁷, and changes in facial behavior could be motivated by situational demands rather than emotional impulses (e.g., smiling to be polite). Further research is necessary to inform broader questions about the associations between measures of emotion and the emotional content itself."

Consistency in Coding: Another concern relates to the use of different automatic coding software across various databases. This variability in the software used to code facial expressions and vocal expressions could undermine the consistency of these measurements. To ensure greater reliability and validity in the data, I recommend that the authors use a single, standardized coding software across all datasets to improve measurement consistency.

We thank the reviewer for raising this important concern about consistency across different facial and vocal coding software. Despite the assessment of similar parameters (e.g., F0, facial cues) in Datasets 2 and 3, we recognize that there were differences across the automated coding procedures used in these datasets. For example, Dataset 2 used pre-existing software programs (e.g., Emotient), whereas Dataset 3 tended to use novel machine learning models to extract facial and vocal cues. The software in Dataset 2 also extracted cues at every 0.5 second of the videos, whereas the models in Dataset 3 extracted cues at every 1 second. Accordingly, we have revised the limitations subsection to acknowledge that the use of consistent coding software for Datasets 2 and 3 could have reduced any inconsistencies in the measurement of facial and vocal cues.

Discussion

(pp. 21-22) **"Moreover, the two datasets in the present work which included automated measurements of facial and vocal cues (Datasets 2 and 3) used different software programs, which could have exacerbated variability in measurement across the two contexts.** Therefore, future research with naturalistic data may wish to **use a standardized** machine learning approach **and complement such an approach** with human-coded behavior to provide a more complete picture of facial and vocal cues which may relate to language approaches to studying emotion."

Thank you for your time and effort throughout the review process. We look forward to seeing this work published in *Communications Psychology*.

Sincerely,
Shaina Munin
Desmond Ong
Sydney Okland
Gili Freedman
Jennifer Beer